# Evaluating Alternative Correction Methods for Multiple Comparison in Functional Neuroimaging Research

**DOI:** 10.3390/brainsci9080198

**Published:** 2019-08-12

**Authors:** Hyemin Han, Andrea L. Glenn, Kelsie J. Dawson

**Affiliations:** 1Educational Psychology Program, University of Alabama, Tuscaloosa, AL 35487, USA; 2Center for the Prevention of Youth Behavior Problems, University of Alabama, Tuscaloosa, AL 35487, USA

**Keywords:** fMRI, multiple comparison correction, statistical non-parametric mapping, 3DClustSim, threshold-free cluster enhancement

## Abstract

A significant challenge for fMRI research is statistically controlling for false positives without omitting true effects. Although a number of traditional methods for multiple comparison correction exist, several alternative tools have been developed that do not rely on strict parametric assumptions, but instead implement alternative methods to correct for multiple comparisons. In this study, we evaluated three of these methods, Statistical non-Parametric Mapping (SnPM), 3DClustSim, and Threshold Free Cluster Enhancement (TFCE), by examining which method produced the most consistent outcomes even when spatially-autocorrelated noise was added to the original images. We assessed the false alarm rate and hit rate of each method after noise was applied to the original images.

## 1. Introduction

Correcting for multiple comparisons to avoid false positives in fMRI studies has been a significant issue in the field of neuroscience. Because the analysis of fMRI data involves comparisons between more than a hundred thousand brain voxels, failing to control the false positive rate could result in misleading findings [1]. For example, let us assume that we intend to conduct a *t*-test on fMRI images that consist of 91×109×91 (standard MNI template) = 902,629 voxels in each image. If we do 902,629 tests on the same image and employ the default *p*-value threshold, *p* < 0.05, without correction, the likelihood to encounter Type I error at least once during statistical analysis becomes $1-{\left(1-0.05\right)}^{902,629}\approx 1$, which is way greater than the initially-intended *p*-value, 0.05. In this situation, if we naively apply the default threshold without any correction, then we are likely to encounter nearly 902,629 × 0.05 ≈ 45,413 errors [2]. To address this issue, analysis tools widely used by researchers (e.g., Statistical Parametric Mapping (SPM), the FMRIBSoftware Library (FSL), Analysis of Functional NeuroImages (AFNI)), employ various statistical methods. For example, the Family-Wise Error (FWE) correction based on Random Field Theory (RFT) or Bonferroni’s method, which controls the likelihood of false positives in analysis, have been implemented in analysis software [2]. The False Discovery Rate (FDR) correction method [3] has also been used as a measure that controls for false positives and is more sensitive and less likely to produce Type II error. These correction methods are implemented in the tools as a part of default analysis procedures. The tools automatically calculate the corrected threshold based on the number of tested voxels and that of voxels reporting significant activity before correction while analyzing fMRI image files.

Although the aforementioned correction methods have attempted to address the issue of inflated false positives in fMRI research and have been widely used in the field, there are debates regarding the performance of these tools [4,5,6]. Some argue that these tools might also inflate false positives [7]. For instance, although users expect that tools correct the actual *p*-value to 0.05, so that the value becomes identical to the intended nominal *p*-value, 0.05, the tools are likely to produce an actual *p*-value > 0.05, which is erroneous and may lead to incorrect analysis results. Eklund et al. [7] argued that such an issue can occur while analyzing fMRI images with ordinary correction methods because the methods seem to rely on problematic statistical assumptions related to the noise and spatial smoothness.

Inflated false positives is a particularly challenging issue in the field of cognitive neuroscience, particularly that which deals with higher order cognition, such as social cognition [8]. Scholars in the field often examine the neural correlates of psychological processes that are not well defined and may be less easily observable than, for example, sensory-motor processes in which there is often less variability between trials and between subjects [9]. Furthermore, the measurement of small effect sizes and experiments that often have low statistical power are major issues in the fields [10]. Particularly for cognitive neuroscience that addresses higher order cognition, finding a balance between Type I and Type II error has been a challenge. Establishing a correction method for multiple comparisons that is more likely to achieve this balance may help to improve our confidence in the results of individual studies.

In this situation, we may consider alternative thresholding methods as potential solutions. Researchers have developed tools implementing correction methods that are not based on parametric statistics (e.g., Statistical NonParametric Mapping (SnPM) [11], Threshold Free Cluster Enhancement (TFCE) [12], 3DClustSim [13,14]). Compared with the traditional correction methods, the aforementioned tools tend to rely on fewer assumptions related to the empirical null hypothesis [15], so they are less likely to cause statistical issues that are associated with parametric assumptions that constitute the basis of classical analysis methods. These new thresholding methods are adopted in the widely-used analysis tools as a function (TFCE in FSL and 3DClustSim in AFNI) or toolbox (SnPM in SPM).

First, SnPM is implemented as a toolbox for SPM using MATLAB. SnPM utilizes permutations that correct for multiple comparisons with fewer parametric assumptions pertaining to the empirical null hypothesis compared to parametric-based methods [15]. SnPM utilizes the general linear model method to create images containing pseudo *t*-scores; the created images are tested for significance using the non-parametric multiple comparison method with permutations [16]. This tool supports both cluster-wise and voxel-wise inference based on FWE, as well as FDR [16].

Second, 3DClustSim is implemented in AFNI. 3DClustSim employs Monte Carlo simulation of Gaussian noise to estimate the probability of false positive clusters [13]. 3DClustSim determines “a cluster-size threshold for a given voxel-wise p-value threshold, such that the probability of anything surviving the dual thresholds is at some given level [17].” The original version of 3DClustSim assumed a Gaussian-shaped autocorrelation function, but recent studies have shown that this assumption might inflate false positives [4,7,13]. Thus, the recently-updated version fixed this issue and provided a new option to use the independently-calculated spherical autocorrelation function parameters [4,13].

Third, TFCE is currently implemented in the randomize function in FSL [18]. TFCE re-weights the resultant *t* or F statistics of every voxel based on the statistics of adjacent voxels; the raw statistics at each point are re-weighted and replaced with the weighted average of the integral (or sum) of the statistics of voxels below that point [12,19]. As a result, in TFCE “the voxel-wise values represent the amount of cluster-like local spatial support” (Smith and Nichols, 2009, p. 83) without requesting users to set a specific cluster-wise threshold. Recently, equitable thresholding and clustering, which implements similar functionality, has been developed for use in AFNI [20].

Although three aforementioned methods can be considered as alternative thresholding methods, there have not been many previous studies that examined whether the methods perform properly as intended in realistic situations. Thus, it is necessary to evaluate these tools to figure out which may be most appropriate for reducing false positives. In the present study, we intended to approach the issue from the end users’ perspective. We focused on concrete datasets that were collected from a limited number of participants, which have been widely recruited in cognitive neuroscience studies, with default setting values provided by the analysis tools, which are usually employed by ordinary end users.

One of the issues that arises when attempting to evaluate methods for multiple comparison is that it is not possible to determine what the “true” pattern of activity should be, nor is there a gold standard for evaluation. The majority of previous studies evaluating methods for multiple corrections utilized data collected from basic perception and clinical and behavioral experiments or used simulation data as the basis for evaluation [2,7,11]. However, the resultant effect size for these findings was large compared to the effect sizes typically observed in studies of higher order cognition. Furthermore, the “correct” neural activity was more easily predicted than in studies of higher order cognition. Hence, in order to enable the evaluation of these tools for multiple comparison correction within the context of cognitive neuroscience with real datasets, we evaluated false alarm and hit rates when noise was added to the original images. For instance, a previous study conducted by Han and Glenn [8] evaluated thresholding methods implemented in SPM 12 by comparing thresholded outcomes with results from meta-analysis [21].

Because we used real experimental fMRI data, we decided to examine whether each thresholding method could produce consistent outcomes even after noise was added to the original images as a way to evaluate whether each method avoided false positives while maintaining sensitivity. We used a false positive rate and hit rate as our two indicators in order to evaluate the propensity to inflate false positives and sensitivity of each method, respectively. This evaluation method was used in a previous study evaluating classical and Bayesian inference methods that were implemented in SPM12 [22]. We aim to employ the indicators used in this previous study to evaluate the performances of alternative thresholding methods, SnPM, 3DClustSim, and TFCE, in the present study. Assessing the aforementioned consistency would be a possible and practical way to evaluate the performance of the methods without relying on the “true” pattern as a gold standard. Since there have not been any prior studies that compared the alternative thresholding methods with real image datasets, our study is exploratory and does not intend to confirm any a priori hypotheses.

### The Present Study

In the present study, we aimed to evaluate SnPM, 3DClustSim, and TFCE, which are implemented in the three most frequently-used analysis tools, by examining whether they were able to produce consistent results when spatially-autocorrelated noise was added to the images. The comparisons between the results from the analyses of the images with and without noise were performed to examine whether the tested correction methods were able to produce consistent results even in the presence of noise and thus be considered reliable. The aforementioned methods based on permutations were evaluated in the context of cognitive neuroscience experiments.

To perform the evaluation, we reanalyzed two sets of fMRI statistical image files that were available via the internet. First, we reanalyzed statistical contrast images from a moral psychology experiment (the moral psychology fMRI dataset) in which participants read and responded to the moral dilemmas developed by Greene et al. [23] and utilized by Han et al. [24]. Second, we reanalyzed a different dataset of statistical contrast images, the working memory dataset. This dataset was created from a cognitive neuroimaging study that examined the neural correlates of three-back working memory processing [25]. We used a part of this dataset that contained images collected from fifteen participants and reanalyzed them to examine the generalizable neural representation of high-order cognitive control [26]. The acquired functional brain images were reanalyzed and thresholded with the three different permutation-based tools. Then, we added spatially-autocorrelated noise to the original images and performed the thresholding processes with the tools again. We decided to do this because prior research paid attention to the spatially-correlated noises, such as physiological and hardware-related noises, as major concerns in data processing (e.g., [27]). Finally, we compared the results from the analyses of the original images and noise-added images to test whether thresholding methods can produce consistent outcomes even in the presence of noise.

## 2. Materials and Methods

### 2.1. Subjects and Materials

The datasets used in this study was comprised of two fMRI datasets available online, which provided access to statistical images from first-level analyses for: (1) sixteen participants who completed a moral dilemma task and (2) fifteen participants who completed a working memory task. For the present study, we first reanalyzed previously-collected social cognition fMRI data [24] that were acquired while subjects were solving moral dilemmas developed by Greene et al. [23] and Greene et al. [28]. The dilemma set consisted of a total of 60 dilemmas including 22 moral-personal, 18 moral-impersonal, and 20 non-moral dilemmas. A total of sixteen participants were involved in the experiment. The original fMRI images that were acquired during the experimental sessions were preprocessed with SPM 8. Han et al. [24] used the RETROICORand RVHRCORmethods to remove physiological artifacts that originated from respiratory and cardiac activities [29,30]. After removing the artifacts, slice time correction, scan drift correction, motion correction, co-registration normalization (into SPM 8’s standard MNI space (79 × 95 × 68 voxels, 2 × 2 × 2 mm^3^ per voxel)), and spatial smoothing (with Gaussian FWHM = 8 mm) were performed. In the present study, we used contrast images that were created from first-level analysis that compared activity between moral (moral-personal + moral-impersonal) versus non-moral conditions. The moral psychology fMRI dataset is available via NeuroVault (https://neurovault.org/collections/3035/) [31].

Second, we reanalyzed the working memory task dataset with the same thresholding methods. This dataset was initially collected to examine the association between the neural correlates of working memory processing and personality traits [25]. For the present study, we used a reprocessed dataset that was created from the original dataset by a group of neuroimaging researchers who intended to illuminate the generalizability of the neural circuitry underlying higher order cognition [26]. This modified dataset contained contrast images from fifteen participants. DeYoung et al. [25] used SPM 2 for preprocessing and analysis. The collected images were realigned using INRIAlignto correct for movement, normalized to the MNIspace, resampled into 3-mm isotropic voxels, and smoothed with an 8-mm FWHM Gaussian kernel. The contrast used for the reanalysis was “3-back working memory versus baseline.” The dataset is available to the public and was downloaded from a lab repository (https://canlabweb.colorado.edu/files/MFC_Generalizability.tar.gz) [32].

### 2.2. Procedures

#### 2.2.1. fMRI Data Reanalysis

First, we reanalyzed the moral psychology fMRI dataset by conducting second-level analysis of the shared contrast images, which were created from first-level analysis, with SPM 12. According to the description of the moral psychology fMRI study, the original functional images were smoothed using an 8-mm full-width at half-maximum Gaussian kernel [24]. We composed a custom MATLAB code in order to add spatially-autocorrelated noise to each original contrast image for our evaluation. We performed the addition of the noise on a research computing cluster, the UAHPC, to facilitate the required large matrix-related operations. Second, in addition to the aforementioned moral psychology fMRI data, we also analyzed one additional higher order cognition fMRI dataset available in a public lab repository. Because the shared files were contrast image files that were created from first-level analyses, we conducted second-level analysis of the image files for our evaluation. Similar to the moral psychology fMRI dataset, the functional images in this dataset were also smoothed using an 8-mm full-width at half-maximum Gaussian kernel. We also performed the same procedures to add spatially-autocorrelated noise to the original contrast images.

We employed the three tools (i.e., SnPM, 3DClustSim, and TFCE) for multiple comparison correction during the second-level analysis. When employing the correction methods, we attempted to follow the guidelines (e.g., threshold setting) presented in official manuals or tutorials because they reflect the settings usually employed by users [33], and we intended to evaluate the performance of each tool from the perspective of users. As a result, we used *p* < 0.001 for the cluster-forming threshold and *p* < 0.05 (FWE for SPM and SnPM, and 3DClustSim) for the cluster-wise threshold. In the case of TFCE, which did not require any cluster-wise threshold setting, we used *p* < 0.05 for thresholding. For all permutation-based methods, we performed permutations 5000 times.

To begin with, we first ran SnPM 13 implemented in SPM’s toolbox [11]. In addition, we applied AFNI’s 3DClustSim to calculate the cluster size threshold. 3DClustSim was performed with two different options. To test 3DClustSim, we first ran 3DClustSim with the mask image created from SPM without activating the autocorrelation function (acf) option implemented in the updated AFNI, which allows the estimation of non-Gaussian acf to improve the reduction of the false positive discovery rate [13]. Second, we estimated parameters for the acf option by running the 3dFWHMxfunction with the residual image created by SPM and performed 3DClustSim with the acf option. Finally, we performed TFCE implemented in FSL [18]. For all second-level analyses using the aforementioned correction methods, we used the contrast image created from the first-level analysis.

#### 2.2.2. Noise-Added Image Creation for Evaluation

We evaluated the consistency of each method based on results from thresholding procedures explained in the previous section. We investigated which voxels survived each thresholding method by analyzing the original contrast images. Binary images containing surviving voxels were created after performing each thresholding method. Then, we added spatially-autocorrelated noise to the original contrast images. We utilized the spatially-autocorrelated noise since the noise in fMRI images, such as the physiological noise, is likely to be spatially correlated according to previous research [7,34]. Because the original fMRI images were smoothed before statistical analysis and the noise in a specific voxel is likely to be correlated with the noise in proximal voxels due to the presence of the global signal and its segregation based on the type of a specific tissue (e.g., gray matter, white matter, ventricle), the noise in fMRI images is likely to be spatially autocorrelated [35,36]. The spatially-correlated noise would be a significant confounding factor in second-level analysis, so it would be a major issue that should be addressed within the context of second-level analysis [37]. Hence, we intended to use the spatially-autocorrelated noise, which has been regarded as a major confounding factor, in our analysis.

First, we created the spatial autocorrelation matrix with the residual time series image file (ResMS) that was created from the analysis of the original images. The spatial autocorrelation between two voxels can be calculated as follows:$$\rho =a\times exp\left(-\frac{r^{2}}{2\times b^{2}}\right)+\left(1-a\right)\times exp\left(-\frac{r}{c}\right)$$
where *r* is the distance between those two voxels. The parameters, *a*, *b*, and *c* were calculated with AFNI’s 3dFWHMx and the created ResMS image file [38,39]. Once the ResMS image file was entered into the 3dFWHMx tool, it automatically calculated three parameters presented in the prior equation, *a*, *b*, and *c*. Second, we calculated autocorrelation coefficients within cubes with a size of 30 × 30 × 30 voxels; this cube size was determined in the consideration of the available memory size at the UAHPC research computing system. Third, we calculated spatially-autocorrelated white noise with a customized MATLAB script titled *correlatedGaussianNoise* that is available via MathWorks File Exchange [40] (further details regarding the noise addition algorithm is explained in and MATLAB codes are shared via https://osf.io/jucgx/ [41]). The MATLAB script requires the aforementioned three parameters, *a*, *b*, and *c*, as inputs to generate the noise. Those three parameter values can be altered according to the result of 3dFWHMx by modifying lines in the MATLAB script that declare the parameter values. Fourth, the magnitude of spatially-autocorrelated noise in each voxel was adjusted to 100% of the mean contrast strength of each contrast image. Once we created the noise-added images, we performed the thresholding processes with the noise-added contrast images, examined surviving voxels, and created binary images once again. The binary images created with the original and noise-added contrast images were used for consistency evaluation. In the case of the analysis of images with random noise, the evaluation processes were performed ten times per dataset.

#### 2.2.3. Statistical Assessment of Consistency

We used two consistency indicators to evaluate whether each thresholding method can produce a consistent outcome even after adding the spatially-autocorrelated noise to the images. The two consistency indicators, false alarm, and hit rates were calculated for each dataset after adding noise to the original image files in order to assess each method’s consistency. A false alarm rate is defined as “the ratio of voxels marked as active from the analysis of noise-added images but as inactive from the analysis of original images to voxels marked as active from the analysis of noise-added images” [22]. The hit rate is defined as “the ratio of voxels marked as active from both analyses to voxels marked as active from the analysis of the original images” [22]. Although we did not use simulated images containing true positives, the false alarm rate and hit rate can be similarly understood as an FDR and sensitivity, respectively [2,22,33]. As mentioned in the Introduction, we were mainly interested in whether the tested correction methods could produce consistent results even when images contained noise. We used indicators for consistency to also indicate the performance of each correction method. Hence, we decided to employ the aforementioned two indicators for our evaluation. We used a lower false alarm rate and higher hit rate as indicators of better consistency. These two consistency indicators were calculated by comparing the binary images created with the original and noise-added contrast images.

We compared the consistencies between three thresholding methods. Because there were two available options for SnPM (cluster-wise and voxel-wise inference) and 3DClustSim (with and without acf option), a total of five conditions (SnPM cluster-wise, SnPM voxel-wise, 3DClustSim with acf, 3DClustSim without acf, and TFCE) were compared. We conducted an ANOVA by setting the false alarm or hit rate as the dependent variable and the thresholding method as the independent variable. These comparisons were performed for each dataset. In addition to the conventional examination of *p*-values, we examined Bayes Factors (BF) to compare models and test each effect. According to the suggested statistical guidelines, we considered 2 log BF < 2 as evidence “not worth more than a bare mention,” 2 ≤ 2 log BF < 6 as “positive” evidence, 6 ≤ 2 log BF < 10 as “strong” evidence, and 2 log BF > 10 as “very strong” evidence supporting our hypothesis [42,43]. Bayesian inference was utilized to examine directly the strength of evidence supporting our hypothesis instead of using *p*-values and to find the best model predicting outcomes among all possible candidate models. In fact, recent debates about statistical testing have warned about using *p*-values and recommend the utilization of additional testing methods [43,44,45,46]. After conducting the ANOVAs, we conducted classical and Bayesian post-hoc tests to examine inter-condition differences. For the classical post-hoc test, we used Scheffe’s method for correction.

## 3. Results

### 3.1. Thresholding Results with Different Methods

Table 1 demonstrates how many voxels survived after running the different correction tools. The results are depicted in Appendix A.

### 3.2. Comparing Consistency Outcomes between SnPM, 3DClustSim, and TFCE

We compared consistency outcomes between SnPM, 3DClustSim, and TFCE with two datasets. A series of Bayesian ANOVA and classical regression analyses was performed to examine which type of applied thresholding method was significantly associated with differences in false alarm and hit rates. False alarm and hit rates are presented in Figure 1.

The results of classical and Bayesian inference revealed whether the thresholding method type was significantly associated with the false alarm or hit rate. When we examined the moral psychology fMRI data, we found a significant main effect of the thresholding method type on both false alarm, *F* (4, 45) = 69.92, *p* < 0.001, 2 log BF = 76.92, and hit rate, *F* (4, 45) = 28.09, *p* < 0.001, 2 log BF = 53.53. Results of classical and Bayesian post-hoc tests demonstrated differences between thresholding methods. In the case of the false alarm rate, TFCE showed a significantly lower false alarm rate, and SnPM voxel-wise inference showed a significantly higher false alarm rate compared with other methods. All these inter-condition differences were significant at *p* < 0.05 (with Scheffe’s method) and 2 log BF > 6. In the case of the hit rate, SnPM voxel-wise inference showed a significantly lower hit rate compared with other methods. The same patterns were found from the working memory fMRI data. The main effect of the thresholding type was significant for both false alarm, *F* (4, 45) = 99.21, *p* < 0.001, 2 log BF = 120.86, and hit rates, *F* (4,45) = 9.80, *p* < 0.001, 2 log bf = 40.88. The results of post-hoc tests with this dataset were identical to those with the moral psychology fMRI dataset.

## 4. Discussion

We evaluated tools implementing correction for multiple comparisons (i.e., SnPM, TFCE, 3DClustSim), which utilize permutations rather than relying on parametric assumptions, within the context of social and cognitive neuroscience research. We used data from a moral psychology fMRI experiment and a working memory fMRI experiment for our evaluation. In order to determine which method showed the best consistency, we compared outcomes for contrast images that did and did not contain spatially-autocorrelated noise. To quantify the consistency of each correction method, we calculated the false alarm and hit rates. In general, TFCE produced the best false alarm rate, while the hit rate was not significantly different across the methods except SnPM voxel-wise inference, which showed a significantly worse hit rate.

Results from thresholding processes in terms of the number of surviving voxels provided information regarding the conservativeness (or selectivity) and leniency (or sensitivity) of each thresholding method (see Table 1 for further details). First, in general, the voxel-wise inference method, SnPM voxel-wise inference in particular, produced the least number of surviving voxels compared with other methods. This result is in line with the conservative nature of SnPM voxel-wise inference that has been reported in previous research (for a review, see [15]). On the other hand, TFCE showed the highest sensitivity given that it produced the greatest number of surviving voxels in our analyses. This result is consistent with the prior research that compared TFCE and other methods and reported its greater sensitivity compared with voxel-based and cluster-based thresholding methods [12]. Hence, in terms of the selectivity and sensitivity of different thresholding methods, our study was able to reproduce similar trends that have been found in previous studies.

In general, TFCE produced a significantly lower false alarm rate compared with the other methods. Given this finding, TFCE is likely to show greater selectivity, even with noise, compared with other thresholding methods [12,33]. This result is in line with the finding from a previous study that tested the validity of TFCE [47]. This prior study showed that TFCE and cluster mass-based thresholding methods better controlled the FWE rate compared with cluster extent and cluster height-based thresholding methods. In addition, TFCE was less influenced by the signal-to-noise ratio, which is closely associated with the presence and strength of noise, compared with other methods. Thus, it would be plausible that TFCE would show a better false alarm rate, which is perhaps associated with a tendency to indicate false positives, even with the presence of noise compared with other thresholding methods in our study. Our study could provide additional evidence that may support the validity of TFCE in terms of controlling the FWE rate with the analyses of real datasets.

Compared with the other methods, SnPM voxel-wise inference produced a relatively higher false alarm rate and a lower hit rate. First, the resulting higher false alarm rate of SnPM voxel-wise inference can be attributed to its conservativeness. As mentioned earlier, SnPM voxel-wise thresholding is more conservative and likely to produce less surviving voxels compared with other methods in ordinary cases [15]. In the calculation of the false alarm rate in our study, as introduced in the Methods Section, the number of surviving voxels was used as a denominator. Because SnPM voxel-wise inference initially produced fewer surviving voxels, the aforementioned denominator became smaller compared with the cases of other thresholding methods, and the resulting false alarm might have become significantly greater even with small changes in the surviving voxels after the addition of noise. Hence, it might be possible to conclude that SnPM voxel-wise inference is more susceptible to noise given its higher false alarm rate.

Second, the method of thresholding for SnPM voxel-wise inference might contribute to its lower hit rate compared with other methods. Unlike other methods, such as cluster-wise thresholding methods that employ the cluster extent threshold and TFCE that examines the strengths of neighbor voxels, SnPM voxel-wise inference does thresholding only based on each individual voxel’s contrast value [15]. When noise is added to the original images, it is likely to increase the standard error value in each voxel during the *t*-test [48], so the increased standard error value decreases the possibility of surviving thresholding, particularly among voxels with pseudo-*t* values that are close to the initial pseudo-*t* thresholding value calculated before the noise addition. Hence, the aforementioned impact of the added noise on the increased standard error in voxels might more strongly influence the hit rate when SnPM voxel-wise inference, which does the thresholding at each individual voxel, is performed. This resulted in a significantly lower hit rate of SnPM voxel-wise inference compared with other methods.

Given the above, TFCE seemed to show relatively better consistency compared with other methods, and users may consider employing this method as their primary choice. However, in the present study, since we were only interested in evaluating the consistency of the methods that can be regarded as an indicator of reliability, the results from the present study do not necessarily imply that TFCE is the best thresholding method in general. Statistically, a good method should possess both good reliability and validity [49]. Reliability is associated with whether the method can produce a certain outcome consistently across situations, which was examined in the present study. Validity is about whether the method can appropriately produce the intended outcome. We did not examine the validity of the thresholding methods, so users should be aware that the results from our study do not necessarily imply that TFCE possesses better validity compared with other methods, although TFCE possesses better consistency (or reliability).

### Limitations

There are several limitations in our study that may warrant further studies. First, although we evaluated the performance of three thresholding methods by comparing outcomes with original and noise-added contrast images, we did not evaluate such performance while assuming true positives, as done in previous simulation-based studies (e.g., Eklund et al. [7]). Because we intended to assess consistency with real datasets while applying thresholding methods and options that were recommended to end users by guidelines, it was necessary to utilize the noise-applied evaluative process instead of the simulation-applied evaluative process based on true positives. Hence, future studies should address the aforementioned issue by employing diverse analysis methods, including simulation-based evaluation and analyzing additional datasets.

Second, although we estimated spatially-autocorrelated noise using ResMS images, our approach is not an ideal method for reproducing realistic noise. This is because the ResMS images were created from first-level and second-level analyses and do not contain any further details regarding time-related information, such as BOLDsignals in fMRI time series that can be used to estimate possible temporal autocorrelation of noise. Because we aimed to focus on thresholding methods used for second-level analysis, we employed statistical images created from first-level analysis that did not possess any temporal information regarding BOLD signals. Thus, to address this issue, future studies may need to reanalyze raw image files, instead of statistical images that were used in our study.

Third, we could only take into account random noise since only the statistical images that contained results from first-level analysis were available for the public. We could not take into account diverse types of noise, such as non-random systematic noise that can be produced by head and body movement or respiratory and cardiac activities [50]. Future studies should address this limitation by generating the non-random systematic noise with additional information, such as information regarding movement and physiological activities, if available.

Fourth, although it would be ideal to calculate the spatially-autocorrelated noise in the whole brain to generate the realistic noise, due to the computational complexity and resource availability on the UAHPC, we could only calculate the noise within 30 × 30 × 30 voxels. Because the cube size can influence the nature of the generated spatially-autocorrelated noise, the limited cube size that was employed in the present study could be a limitation. In the current setting, our MATLAB script required nearly 40 GB of memory. To calculate the noise in the whole brain, approximately 1 TB (= 1024 GB) of memory is required according to our estimation. It would be necessary to do the whole-brain noise calculation to simulate the more realistic noise once we have sufficient computational power and resources.

Fifth, because we were mainly interested in finding a way to evaluate different alternative thresholding methods, we only analyzed two datasets. Future studies should conduct evaluations of additional datasets for better generalization.

## 5. Conclusions

We evaluated different tools implementing permutation-based methods for correcting multiple comparisons, SnPM, 3DClustSim, and TFCE, within the contexts of cognitive neuroscience fMRI research. The false alarm and hit rates were assessed by comparing outcomes from the original images and images with spatially-autocorrelated noise based on the residual mean squared images. The evaluation processes were performed with two different datasets, moral psychology fMRI and working memory fMRI, that have different types of clusters of surviving voxels. Our study showed that the performance in terms of consistency was significantly different across multiple thresholding methods. Particularly, TFCE showed a relatively better false alarm rate compared with other methods, while the hit rate was not significantly different across the methods except SnPM voxel-wise inference.

## Figures and Tables

**Figure 1 brainsci-09-00198-f001:**
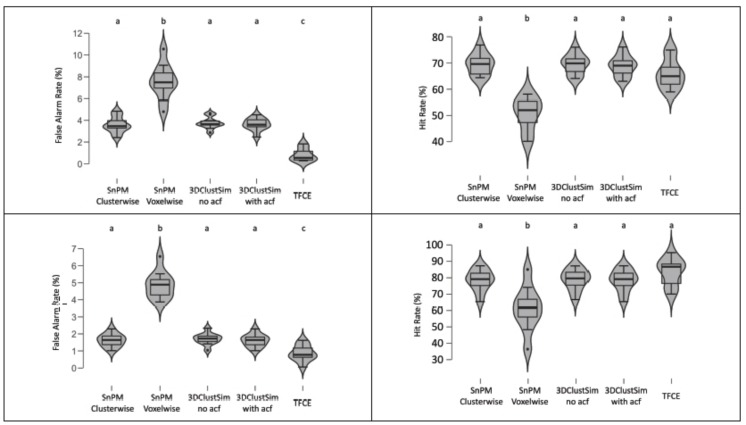
Violin plots of calculated false alarm and hit rates in five different conditions. (**Top-left**) Moral psychology fMRI false alarm rate. (**Top-right**) Moral psychology fMRI hit rate. (**Bottom-left**) Working memory fMRI false alarm rate. (Bottom-right) Working memory fMRI hit rate. Conditions that did not produce a significantly different false alarm or hit rate compared with each other are represented with the same letter. The *p*-value threshold, *p* < 0.05, and Bayes factor threshold, 2 log BF = 2, were applied.

**Table 1 brainsci-09-00198-t001:** Surviving voxel number for each correction method.

Correction Method		Number of Surviving Voxels
Moral psychology fMRI data		
SnPM voxel-wise *p* < 0.05 (FWE)		860
SnPM cluster-wise *p* < 0.05 (FWE)	cluster-forming *p* < 0.001	14,945
3DClustSim cluster-wise *p* < 0.05	cluster-forming *p* < 0.001	15,659
3DClustSim (acf) cluster-wise *p* < 0.05	cluster-forming *p* < 0.001	15,273
TFCE	corrected *p* < 0.05	32,272
Working memory fMRI data		
SnPM voxel-wise *p* < 0.05 (FWE)		820
SnPM cluster-wise *p* < 0.05 (FWE)	cluster-forming *p* < 0.001	15,291
3DClustSim cluster-wise *p* < 0.05	cluster-forming *p* < 0.001	15,426
3DClustSim (acf) cluster-wise *p* < 0.05	cluster-forming *p* < 0.001	15,291
TFCE	corrected *p* < 0.05	20,403

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
