# Peer review of "Evaluating Alternative Correction Methods for Multiple Comparison in Functional Neuroimaging Research"

_brainsci, 2019, doi:10.3390/brainsci9080198_

Round 1
Reviewer 1 Report
I thank the editors for the opportunity to review this article titled “Evaluating Alternative Correction Methods for Multiple Comparison in Functional Neuroimaging Research” for publication in MDPI Brain sciences – Neuroimaging. In this article, the authors report a challenging and well-designed study into the comparative efficacy of three of the more novel methods for multiple comparisons correction of fMRI data. The article reviews a variety of existing methods for multiple comparison correction of fMRI data and addresses the need for such methods. Then three such methods are used on two existing, previously published fMRI datasets. The methods are compared for their ability to give consistent thresholding even when spatially autocorrelated noise was introduced. This was achieved by assessing the false alarm rate and hit rate of each multiple comparison correction method after the noise was applied to the original images.
The authors provide a thorough account of the methods used in the study and I have relatively few concerns about the article. I do have a number of suggestions to improve the clarity of the article and the completeness of the discussion, thus justifying the impact of the article in the field.
Introduction:
The first paragraph of the introduction (lines 13-22) is a nice succinct introduction into the methods of multiple comparison correction typically used in fMRI. I feel it is rather too brief and would benefit from just a little more expansion on how each method is implemented. This would then assist with the next issue.
Line 54 refers to “the issue of inflated false positives in fMRI research”. This has not been introduced yet. The authors go on in the next paragraph to discuss the potential consequences of inflated false positives, and the impact on the field. But there is no text explicitly introducing and justifying the existence of an issue in the first place. Please describe fully the problem and give published citations to back it up.
Further, in line 55, the authors state that some tools have been shown to “also inflate false positives” – please expand on this by explaining through what mechanism this occurs prior to just citing Eklund et al.
Do the authors, or the current literature, form any hypotheses about which method might be more effective at preventing false positives, false negatives, or more consistent following the application of noise?
Line 15 should read “more than a hundred thousand brain voxels” – the “a” is absent.
Methods:
Line 124 through 138: I find the description of the data preprocessing lacking. The authors explicitly state that the working memory fMRI dataset was already preprocessed prior to its use in the present study. This is not stated explicitly for the moral dilemmas study. However, in the discussion, it is inferred that statistical images from first level analyses were used as opposed to pre-processed fMRI timecourses. Please explicitly state whether pre-processed timecourses were used or whether statistical images from first level analysis were used for each of the two datasets. Also a summary of the preprocessing methods that had previously been applied by the original publishers of the datasets would be useful at this stage so that the reader doesn’t have to keep referring back to multiple cited papers to get this information.
The concept of spatially autocorrelated noise is introduced in line 144. Please briefly expand to provide a short definition of spatially autocorrelated noise and how it is generated. Were any parameters needed to be specified for the running of the Matlab file at all? Please either state or provide them here.
As a personal preference on ease of navigation of the article, I would split section 2.2.2 Consistency evaluation into two sections, perhaps one covering addition of spatially autocorrelated noise, and a second describing the *statistical* assessment of consistency of the methods.
Results:
Figure 1, in the legend, please add the statistical threshold at which “conditions that did not produce a significantly different false alarm or hit rate” were assessed.
Discussion and conclusion:
I do not feel that the authors are making any conclusions from the analyses in the article. Please at the beginning of the Discussion section, summarise the findings from the results section. Further, please conclude in the Conclusion section, which multiple comparison method(s) were preferred or recommended against, and any caveats to that recommendation. It would be nice to have a reference to this overarching result in the abstract as well.
Note on language:
As the language in this article is excellent, I do not want to make heady-handed comments. There are just a few instances where some ambiguity creeps in through use of an atypical emphasis within a description. E.g. line 129-130 “we repeated the same evaluation process with one additional dataset” sounds like you acquired one additional participant yourselves or something. I would recommend prefacing the methods section with something along the lines of “the data used in this study comprised two fMRI datasets available online, and accessed as statistical images from first level analyses: (1) sixteen participants who completed a moral dilemmas task and (2) fifteen participants who completed a working memory task”.
Some parts of the article seem to outline a method that was used, then fully describe a different method, then outline a third method, then fully describe the first method, etc. in a leap-frogging order. This was sometimes confusing to the reader, so I recommend just reading through the article for clarity of information.
Author Response
Introduction:
The first paragraph of the introduction (lines 13-22) is a nice succinct introduction into the methods of multiple comparison correction typically used in fMRI. I feel it is rather too brief and would benefit from just a little more expansion on how each method is implemented. This would then assist with the next issue.
Thank you very for your constructive comments. In the revised manuscript, we provided further details regarding the correction methods and how they are implemented in widely-used analysis tools.
These correction methods are implemented in the tools as a part of default analysis procedures. The tools automatically calculate the corrected threshold based on the number of tested voxels and that of voxels reporting significant activity before correction while analyzing fMRI image files. (pp. 1-2)
Line 54 refers to “the issue of inflated false positives in fMRI research”. This has not been introduced yet. The authors go on in the next paragraph to discuss the potential consequences of inflated false positives, and the impact on the field. But there is no text explicitly introducing and justifying the existence of an issue in the first place. Please describe fully the problem and give published citations to back it up.
We appreciate your comments regarding the inflated false positives. We explained the nature of the problem in the revised manuscript.
For example, let us assume that we intend to conduct t-test on fMRI images that consist of 91 x 109 x 91 (standard MNI template) = 902,629 voxels in each image. If we do 902,629 tests on the same image and employ the default p-value threshold, p < .05, without correction, the likelihood to encounter Type I error at least once during statistical analysis becomes 1? (1 ? .05)902,629 1, which is way greater than the initially intended p-value, .05. In this situation, if we naively apply the default threshold without any correction, then we are likely to encounter nearly 902, 629 .05 45, 413 errors [2]. (p. 1)
Further, in line 55, the authors state that some tools have been shown to “also inflate false positives” – please expand on this by explaining through what mechanism this occurs prior to just citing Eklund et al.
Thanks a lot for your invaluable comment. We explained further details regarding the issue in the revised manuscript.
Although the aforementioned correction methods have attempted to address the issue of inflated false positives in fMRI research and have been widely used in the field, there are debates regarding the performance of these tools [4–6]. Some argue that these tools might also inflate false positives [7]. For instance, although users expect that tools correct the actual p-value to .05 so that the value becomes identical to the intended nominal p-value, .05, the tools are likely to produce an actual p-value > .05, which is erronous and may lead to incorrect analysis results. Eklund et al. [7] argued that such an issue can occur while analyzing fMRI images with ordinary correction methods because the methods seem to rely on problematic statistical assumptions related to the noise and spatial smoothness. (p. 2)
Do the authors, or the current literature, form any hypotheses about which method might be more effective at preventing false positives, false negatives, or more consistent following the application of noise?
We appreciate your suggestion regarding the presentation of hypotheses. Because our study is exploratory, we did not start with any a prior hypothesis. We specified that in the revised manuscript.
Since there have not been any prior studies that compared the alternative thresholding methods with real image datasets, our study was exploratory and did not intend to confirm any a priori hypotheses. (p. 4)
Line 15 should read “more than a hundred thousand brain voxels” – the “a” is absent.
Thank you for your correction. We corrected the typo. In addition, we have our manuscript reviewed and edited by a naive English speaker to correct any typos and errors.
Methods:
Line 124 through 138: I find the description of the data preprocessing lacking. The authors explicitly state that the working memory fMRI dataset was already preprocessed prior to its use in the present study. This is not stated explicitly for the moral dilemmas study. However, in the discussion, it is inferred that statistical images from first level analyses were used as opposed to pre-processed fMRI timecourses. Please explicitly state whether pre-processed timecourses were used or whether statistical images from first level analysis were used for each of the two datasets. Also a summary of the preprocessing methods that had previously been applied by the original publishers of the datasets would be useful at this stage so that the reader doesn’t have to keep referring back to multiple cited papers to get this information.
We sincerely appreciate your suggestion. We explained the pre-processing processes in the revised manuscript. Also, we specified that the contrast images that were created from first-level analysis were analyzed in our study.
The original fMRI images that were acquired during the experimental sessions were preprocessed with SPM 8. Han et al. [24] used RETROICOR and RVHRCOR methods to remove physiological artifacts that originated from respiratory and cardiac activities [29,30]. After removing the artifacts, slice time correction, scan drift correction, motion correction, co-registrationn normalization (into SPM 8’s standard MNI space (79 x 95 x 68 voxels, 2 x 2 x 2 mm3 per voxel)), and spatial smoothing (with Gaussian FWHM = 8 mm) were performed. In the present study, we used contrast images that were created from first-level analysis that compared activity between moral (moral-personal + moral-impersonal) versus non-moral conditions.
...
DeYoung et al. [25] used SPM 2 for preprocessing and analysis. The collected images were realigned using INRIAlign to correct for movement, normalized to the MNI space, resampled into 3 mm isotropic voxels, and smoothed with an 8 mm FWHM Gaussian kernel. The contrast used for the reanalysis was "3-back working memory versus baseline.". (pp. 4-5)
The concept of spatially autocorrelated noise is introduced in line 144. Please briefly expand to provide a short definition of spatially autocorrelated noise and how it is generated. Were any parameters needed to be specified for the running of the Matlab file at all? Please either state or provide them here.
Thank you very much for your comments regarding the spatially autocorrelated image. We explained further details regarding the nature of the noise and how we generated the noise in our study.
We utilized the spatially-autocorrelated noise since the noise in fMRI images, such as the physiological noise, is likely to be spatially correlated according to previous research[7,32]. Because original fMRI images are smoothed before statistical analysis and the noise in a specific voxel is likely to be correlated with the noise in proximal voxels due to the presence of the global signal and its segregation based on the type of a specific tissue (e.g., gray matter, white matter, ventricle), the noise in fMRI images is likely to be spatially autocorreated[33,34]. The spatially correlated noise would be a significant confounding factor in second-level analysis, so it would be a major issue that should be addressed within the context of second-level analysis[35]. Hence, we intended to use the spatially-autocorrelated noise, which has been regarded as a major confounding factor, in our analysis.
...
The MATLAB script requires the aforementioned three parameters, a, b, and c, as inputs to generate the noise. Those three parameter values can be altered according to the result of 3dFWHMx by modifying lines in the MATLAB script that declare the parameter values. (p.6)
As a personal preference on ease of navigation of the article, I would split section 2.2.2 Consistency evaluation into two sections, perhaps one covering addition of spatially autocorrelated noise, and a second describing the *statistical* assessment of consistency of the methods.
We sincerely appreciate your suggestion about the subsections. In the revised manuscript, we split the subsection accordingly.
Results:
Figure 1, in the legend, please add the statistical threshold at which “conditions that did not produce a significantly different false alarm or hit rate” were assessed.
Thank you very much for your suggestion. We added a description about which threshold value (p < .05) was used.
Discussion and conclusion:
I do not feel that the authors are making any conclusions from the analyses in the article. Please at the beginning of the Discussion section, summarise the findings from the results section. Further, please conclude in the Conclusion section, which multiple comparison method(s) were preferred or recommended against, and any caveats to that recommendation. It would be nice to have a reference to this overarching result in the abstract as well.
We appreciate your suggestion regarding the discussion section. We added some concluding remarks at the end of the discussion and conclusion sections. In addition, at the beginning of the section, we briefly summarized the findings from our study.
In general, TFCE produced the best false alarm rate while the hitrate was not significant different across the methods except SnPM voxelwise inference, which showedthe significantly worse hit rate. (p. 8)
...
Given these, TFCE seems to show relatively better consistency compared with other methods, and users may consider employing this method as their primary choice. However, in the present study, since we were only interested in evaluating the consistency of the methods that can be regarded as an indicator of reliability, the results from the present study do not necessarily imply that TFCE is the best thresholding method in general. Statistically, a good method should possess both good reliability and validity [46]. Reliability is associated with whether the method can produce a certain outcome consistently across situations, which was examined in the present study. Validity is about whether the method can appropriately produce the intended outcome. We did not examine the validity of the thresholding methods, so users should be aware that the results from our study do not necessarily imply that TFCE possesses the better (p. 10)
...
Particularly, TFCE showed a relatively better false alarm rate compared with other methods, while the hit rate was not significantly different across the methods except SnPM voxelwise inference. (p. 11)
Note on language:
As the language in this article is excellent, I do not want to make heady-handed comments. There are just a few instances where some ambiguity creeps in through use of an atypical emphasis within a description. E.g. line 129-130 “we repeated the same evaluation process with one additional dataset” sounds like you acquired one additional participant yourselves or something. I would recommend prefacing the methods section with something along the lines of “the data used in this study comprised two fMRI datasets available online, and accessed as statistical images from first level analyses: (1) sixteen participants who completed a moral dilemmas task and (2) fifteen participants who completed a working memory task”.
Thanks a lot for your invaluable suggestion. We edited the methods section accorindly.
The datasets used in this study comprised two fMRI datasets available online, which provided access to statistical images from first-level analyses for: (1) sixteen participants who completed a moral dilemmas task and (2) fifteen participants who completed a working memory task. (p. 4)
Some parts of the article seem to outline a method that was used, then fully describe a different method, then outline a third method, then fully describe the first method, etc. in a leap-frogging order. This was sometimes confusing to the reader, so I recommend just reading through the article for clarity of information.
We appreciate your comment regarding the organization of the manuscript. We modified the organization in the revised manuscript.
Reviewer 2 Report
In this work, the authors evaluated different tools implementing permutation-based methods for correcting multiple comparisons, i.e., SnPM, 3DClustSim, and TFCE.
They used two different datasets (moral psychology fMRI and working memory fMRI datasets) to compare the different methods. In particular, the two metrics false alarm and hit rate were assessed by comparing the survived voxels from the original contrast images and the noise-affected images.
Although the study is a useful exploratory analysis to identify the most effective method of correction, there are points to be revised.
Specifically, the authors use spatially autocorrelated noise based on the residual mean square images. More details on the characteristics of this noise and the technique used to generate autocorrelated noise should be provided.
In fact, it is well known that there are several sources of non-thermal noise coherently affecting large brain regions therefore it is appropriate at least to discuss such contributions.
The authors shoud also consider that autocorrelation coefficients are estimated within cubes with a size of 30 x 30 x 30 voxels: althorugh this cubic size was determined in the consideration of the available memory size at the UAHPC research computing system, the size of the cubes affects the autocorrelation.
Author Response
Specifically, the authors use spatially autocorrelated noise based on the residual mean square images. More details on the characteristics of this noise and the technique used to generate autocorrelated noise should be provided.
We appreciate your comment about the explanation of the noise used in our study. We added the description in the revised manuscript.
We utilized the spatially-autocorrelated noise since the noise in fMRI images, such as the physiological noise, is likely to be spatially correlated according to previous research[7,32]. Because original fMRI images are smoothed before statistical analysis and the noise in a specific voxel is likely to be correlated with the noise in proximal voxels due to the presence of the global signal and its segregation based on the type of a specific tissue (e.g., gray matter, white matter, ventricle), the noise in fMRI images is likely to be spatially autocorreated[33,34]. The spatially correlated noise would be a significant confounding factor in second-level analysis, so it would be a major issue that should be addressed within the context of second-level analysis[35]. Hence, we intended to use the spatially-autocorrelated noise, which has been regarded as a major confounding factor, in our analysis.
...
The MATLAB script requires the aforementioned three parameters, a, b, and c, as inputs to generate the noise. Those three parameter values can be altered according to the result of 3dFWHMx by modifying lines in the MATLAB script that declare the parameter values. (p.6)
In fact, it is well known that there are several sources of non-thermal noise coherently affecting large brain regions therefore it is appropriate at least to discuss such contributions.
Thank you very much for your comment regarding the different types of noise. We discussed the point in the limitations section.
Third, we could only take into account random noise since only the statistical images that contained results from first-level analysis were available for the public. We could not take into account diverse types of noise, such as non-random systematic noise that can be produced by head and body movement, or respiratory and cardiac activities [47]. Future studies should address this limitation by generating the non-random systematic noise with additional information, such as information regarding movement and physiological activities, if available. (p. 11)
The authors shoud also consider that autocorrelation coefficients are estimated within cubes with a size of 30 x 30 x 30 voxels: althorugh this cubic size was determined in the consideration of the available memory size at the UAHPC research computing system, the size of the cubes affects the autocorrelation.
Thanks a lot for your comment regarding the generation of the noise. We discussed the point in the limitations section.
Fourth, although it would be ideal to calculate the spatially autocorrelated noise in the whole brain to generate the realistic noise, due to the computational complexity and resource availability on the UAHPC, we could only calculate the noise within 30 x 30 x 30 voxels. Because the cubic size can influence the nature of the generated spatially-autocorrelated noise, the limited cubic size that was employed in the present study could be a limitation. In the current setting, our MATLAB script required nearly 40GB memory. To calculate the noise in the whole brain, approximately 1TB (= 1,024GB) memory is required according to our estimation. It would be necessary to do the whole-brain noise calculation to simulate the more realistic noise once we have a sufficient computational power and resource. (p. 11)